# Procollagen I and III as Prognostic Markers in Patients Treated with Extracorporeal Membrane Oxygenation: A Prospective Observational Study

**DOI:** 10.3390/jcm10163686

**Published:** 2021-08-19

**Authors:** Christoph Boesing, Peter T. Graf, Manfred Thiel, Thomas Luecke, Joerg Krebs

**Affiliations:** Department of Anaesthesiology and Critical Care Medicine, University Medical Centre Mannheim, Medical Faculty Mannheim of the University of Heidelberg, Theodor-Kutzer-Ufer 1-3, 68167 Mannheim, Germany; Christoph.Boesing@umm.de (C.B.); Tobias.Graf@umm.de (P.T.G.); Manfred.Thiel@umm.de (M.T.); thluecke@googlemail.com (T.L.)

**Keywords:** acute respiratory distress syndrome, extracorporeal membrane oxygenation, procollagen I, procollagen III

## Abstract

**Background**: Procollagen peptides have been associated with lung fibroproliferation and poor outcomes in patients with acute respiratory distress syndrome (ARDS). Therefore, serum procollagen concentrations might have prognostic value in ARDS patients treated with extracorporeal membrane oxygenation (ECMO). **Methods:** In a prospective cohort study, serum N-terminal procollagen I-peptide (PINP) and N-terminal procollagen III-peptide (PIIINP) concentrations in twenty-three consecutive patients with severe ARDS treated with ECMO were measured at the time of ECMO initiation and during the course of treatment. The predictive value of PINP and PIIINP at the time of ECMO initiation was tested with a univariable logistic regression and a receiver operating characteristic (ROC) curve analysis. **Results:** Thirteen patients survived to intensive care unit (ICU) discharge. Non-survivors had higher serum PINP and PIIINP concentrations at all points in time during the course of treatment. Serum PIIINP at the day of ECMO initiation showed an odds ratio of 1.37 (95% CI 1.10–1.89, *p* = 0.017) with an area under the receiver operating characteristic (ROC) curve (AUC) of 0.87 (95% CI 0.69–1.00, *p* = 0.0029) for death during the course of treatment. **Conclusions:** PINP and PIIINP concentrations differ between survivors and non-survivors in ARDS treated with ECMO. This exploratory hypothesis generating study suggests an association between PIIINP serum concentrations at ECMO initiation and an unfavorable clinical outcome.

## 1. Introduction

Acute respiratory distress syndrome (ARDS) is a heterogeneous syndrome with a substantial biological variance [1] associated with a high mortality rate of 35–50% [2,3,4] and without any specific pharmacological treatment option [5]. For the most severe cases of ARDS, extracorporeal membrane oxygenation (ECMO) is a treatment option to guarantee oxygenation and facilitate lung protective ventilation [6,7,8].

Procollagen as a precursor of collagen is produced by pulmonary fibroblasts in the extracellular space [9]. The amount of collagen deposition in the fibroproliferative phase of ARDS is determined by the extent of the ventilator associated lung injury and the intensity of the inflammatory stimulus [10,11]. Resulting from enzymatic cleavage of procollagen by specific proteases, amino-terminal procollagen peptides can be used as markers for collagen synthesis [12,13,14,15]. N-terminal procollagen III-peptide (PIIINP) is mainly synthesized in the early course of ARDS whereas N-terminal procollagen I-peptide (PINP) is more prevalent in the later course [16,17]. High concentrations of procollagen III-peptide in serum and bronchoalveolar lavage fluid from ARDS patients have been associated with poor outcome [18,19,20,21,22,23] and elevated concentrations in bronchoalveolar fluid (BAL) during the early course of ARDS are particularly associated with high mortality [22,23,24,25,26]. Because fibroproliferation varies significantly between resolving and unresolving ARDS [27], PINP and PIIINP could have prognostic value in ARDS patients. Furthermore, PINP and PIIINP might guide therapeutic interventions, such as the initiation of ECMO or the use of corticosteroids in the treatment of severe ARDS [28].

We therefore measured the concentrations of serum PINP and PIIINP at several points in time during the course of treatment and evaluated the prognostic value in patients with severe ARDS treated with ECMO.

## 2. Material and Methods

### 2.1. Study Design and Ethics

This prospective observational cohort study was conducted from August 2017 to August 2018 following approval of the local ethics committee (Medizinische Ethikkomission II, University Medical Centre Mannheim, Medical Faculty Mannheim of the University of Heidelberg, Mannheim, registration number 2016-601N-MA) and study registration at the German clinical trials register (DRKS00013967).

### 2.2. Participants

After obtaining written informed consent from each patient or their relatives, 23 consecutive patients with severe ARDS requiring veno-venous extracorporeal membrane oxygenation were studied at the Department of Anaesthesiology and Critical Care Medicine, University Medical Centre Mannheim, Medical Faculty Mannheim of the University of Heidelberg in Mannheim, Germany.

Exclusion criteria were age younger than 18 years, pregnancy (both because of requirements of the ethics committee), end-stage chronic organ failure (prohibiting ECMO therapy) and preexisting pulmonary fibrosis.

Patients with continuous corticosteroid medication as well as patients where the treatment of the underlying disease necessitated a corticosteroid medication were also excluded, as corticosteroids can rapidly reduce the serum concentration of PINP and PIIINP [24].

No patient was excluded from ECMO treatment because of a maximum number of days of ventilation prior to cannulation per the policy of our unit.

Patients were dichotomized in survivors and non-survivors.

### 2.3. Standard Therapy

Patients with severe ARDS according to current definitions were considered for extracorporeal therapy [29]. Apart from patients who received high urgent salvage ECMO because of life threatening hypoxia or were cannulated from the out-of-house retrieval team in another hospital, all patients were treated according to current ARDS recommendations and received an individualized positive end-expiratory pressure (PEEP) management, prone positioning and neuromuscular blocking agents [30]. In case of conservative treatment failure, ECMO therapy was initiated according to standard operating procedures of the department by the attending physician when patients fulfilled criteria published by the extracorporeal life organization guidelines [31].

Patients were initially sedated with midazolam (5 to 15 mg/h) and sufentanil (30 to 40 µg/h) and received neuromuscular blocking agents as indicated. Analgosedation during the course of treatment on intensive care unit (ICU) was managed by the attending physician. The ECMO circuit was set up according to the local protocol with percutaneous placement of a venous drainage cannula using the femoral vein and a return cannula using the jugular vein. Blood flow through the ECMO circuit was adjusted to achieve a PaO_2_ between 65 and 90 mmHg and an arterial oxygen saturation (SaO_2_) over 90%. Sweep gas flow was adjusted to achieve a PaCO_2_ of 35 to 45 mmHg. Initially, all patients were ventilated in the volume-control mode with tidal volumes (V_T_) of 2.5–4 mL/kg idealized body weight (IBW) and respiratory rates (RR) of 10–12/min. Positive end-expiratory pressure (PEEP) was adjusted to the preference of the attending physician during the course of treatment on ICU. End-inspiratory plateau pressure (P_plat_) was recorded during a 5-s inspiratory hold. Driving pressure was calculated as end-inspiratory plateau pressure (P_plat_)—PEEP. Static compliance of the respiratory system (C_rs_) was calculated as C_rs_ = V_T_/(P_plat_—PEEP).

The attending physician carefully evaluated the patients during the ECMO run for their suitability of a spontaneous breathing trial. According to the standard operating procedure of our ECMO unit, this required hemodynamical stability, a marked reduction in the vasopressor dosage and a negative fluid balance for the last 24 h either due to diuretics or continuous renal replacement therapy. If these prerequisites were met, neuromuscular blockade was discontinued, sedative agents reduced and an assisted ventilator mode with lung protective settings established.

The ECMO support was gradually reduced while keeping the PaO_2_ and PaCO_2_ within physiological limits. Vigorous spontaneous breathing and high RR because of a high respiratory drive resulting in injurious driving pressures were treated with modulations of ECMO blood and gas flow, sedatives and neuromuscular blocking agents as indicated by the attending physician [32].

### 2.4. Data Collection

At the time of study inclusion with the beginning of veno-venous ECMO and protective ventilation, anthropomorphic data and duration of mechanical ventilation (MV) prior to inclusion were recorded. The Simplified Acute Physiology Score (SAPS II) [33], Sequential Organ Failure Assessment (SOFA) score [34], Acute Physiology and Chronic Health Evaluation II (APACHE II) score [35], Respiratory ECMO Survival Prediction (RESP) score [36] and Predicting Death for Severe ARDS on veno-venous ECMO (PRESERVE) score [37] were calculated. Length of ICU stay and duration of ECMO support were calculated after the observation period. PINP, PIIINP, white blood cell count (WBC), C-reactive protein (CRP) and procalcitonin (PCT) were measured at the time of ECMO initiation and on day 3, 5 and 10 throughout the period of study. PINP concentrations were measured by electrochemiluminiscence immunoassay (Elecsys total PINP; Roche Diagnostics, Mannheim, Germany) with a normal range for serum PINP between 20.2 and 76.3 µg/L (5th and 95th percentile) in healthy adults. The used assay has an intra-test and inter-test variability equal to or less than 3.2% and 3.7%, respectively. For PIIINP, radioimmunoassay (RIA-gnost PIIIP; Cisbio Bioassays, Codolet, France) was used with a normal range for serum PIIINP between 2.4 and 6.4 µg/L (5th and 95th percentile) in healthy adults and an upper measuring range of 48 µg/L. The used assay has an intra-test variability of 3, 1.7 and 2.6% for PIIINP concentrations of 5.6, 11.2 and 48 µg/L, respectively. The inter-test variability is 7.5, 4.7 and 5.3% for PIIINP concentrations of 2.1, 11.2 and 44.6 µg/L, respectively. The characteristics of the used assays were given by the manufacturers. 

### 2.5. Statistical Analysis

The statistical analysis was performed with Prism Version 8.0.2 (GraphPad Software, San Diego, CA, USA). Categorical variables were compared using Fisher’s exact test and presented as frequency and percentages. For continuous variables, the distribution was tested using the Kolmogorov-Smirnov test. As appropriate, variables were compared using Student’s t-test or the Mann-Whitney U test and expressed as mean (±standard deviation) or median (interquartile range). For the association between PINP and PIIINP and death during ECMO support, we performed a logistic regression to report the predictive value of the marker. The discriminative value of the model was assessed by using the area under the receiver operating characteristic curve (AUC). For the AUC, values of 0.5 indicate no predictive ability, 0.7 to 0.8 is considered acceptable, 0.8 to 0.9 is considered excellent and more than 0.9 is considered outstanding [38]. Calibration and overall goodness of fit was evaluated using the Brier score as proposed in the literature [39]. The Brier score quantifies how close predictions of the model are to the actual outcome and ranges from 0 to 1, whereas a smaller Brier score indicates superior performance of the used model. The fit of the logistic model was further tested using the likelihood ratio test (LRT). Agreement between the predicted probability of the model and the observed probability was graphically assessed by a calibration curve. The best compromise between sensitivity (Se) and specificity (Sp) for PINP and PIIINP was derived from the ROC (Receiver operating characteristics) curve and used to dichotomize the continuous variables. Se, Sp, positive predictive value (PPV), negative predictive value (NPV), likelihood ratios (LR) and diagnostic accuracy were calculated for the chosen thresholds. Diagnostic odds ratio (DOR) was calculated as a single, prevalence-independent indicator of diagnostic performance [40]. DOR values range from zero to infinity, with higher values indicating better discriminatory performance. As the sensitivity and specificity of the test becomes near perfect, the DOR rises steeply [40]. For Se, Sp, PPV and NPV, confidence intervals (CI) were calculated using the Wilson–Brown method. The CI for LR and DOR were determined using the “log method” proposed by Altman et al. [41] and the log DOR with back-transformation [40], respectively. The data are displayed as Box–Whisker Plots using the Tukey method. Missing data was not replaced by using an imputation method. For all statistical tests, *p*-values smaller than 0.05 were regarded as significant.

## 3. Results

Twenty-three patients with severe ARDS treated with veno-venous ECMO were included in the study. Thirteen patients (57%) were successfully weaned and discharged from ICU while ten patients (43%) died. Twelve patients survived to hospital discharge. Table 1 shows the characteristics of the patients, including anthropomorphic data, etiology of ARDS, prediction scores, duration of mechanical ventilation prior to inclusion, length of ICU stay and the duration of ECMO support. 

Prediction scores at the time of ECMO initiation did not differ between survivors and non-survivors. The duration of ECMO support was longer in non-survivors compared to survivors (16.4 ± 8.1 vs. 11.2 ± 2.9 days, *p* = 0.0437). Furthermore, the ICU length of stay was longer in survivors compared to non-survivors (37.0 (32–73) vs. 24.0 (15.8–33.5) days, *p* = 0.0039).

The characteristics of mechanical ventilation, ECMO support and inflammatory parameters are shown in Appendix A. 

### Procollagen I and III

PINP concentrations were higher in non-survivors compared to survivors at the time of ECMO initiation (75.0 (43.8–184.5) vs. 35.0 (24.5–59) µg/L, *p* = 0.0126), day 3 (111.0 (45.5–174.3) vs. 25.5 (22.0–81.5) µg/L, *p* = 0.0067) and day 5 (114.0 (53.0–222.5) vs. 30.0 (19.5–70.0) µg/L, *p* = 0.0071) (Figure 1). The logistical regression for PINP showed a non-significant odds Ratio (OR) of 1.03 (95% CI 1.00–1.07, *p* = 0.09). Therefore, the diagnostic and discriminatory performance of PINP was not further assessed.

Non-survivors had higher PIIINP concentrations compared to survivors at the time of ECMO initiation (18.4 (14.8–28.2) vs. 9.6 (7.6–10.4) µg/L, *p* = 0.0018), on day 3 (21.6 (17.2–44.4) vs. 10.0 (6.6–11.8) µg/L, *p* = 0.0095), day 5 (19.2 (16.8–36.8) vs. 12.0 (6.8–15.6) µg/L, *p* = 0.002) and day 10 (30.4 (17.6–48.9) vs. 14.8 (9.4–27.2) µg/L, *p* = 0.0354) (Figure 2A). The logistical regression for PIIINP at the time of ECMO initiation yielded an OR of 1.37 (95% CI 1.10–1.89, *p* = 0.017) with an AUC of 0.87 (95% CI 0.69–1.00, *p* = 0.0029) (Figure 2B). The Brier score of the model was 0.121 with a LRT of 12.3 (*p* = 0.0005).

Table 2 shows the diagnostic performance including the diagnostic odds ratio of the chosen cut-off for PIIINP at >12.8 µg/L.

Table 3 shows the diagnostic performance of serum N-terminal procollagen III-peptide at cut-offs of >8.4 µg/L, >12.8 µg/L, >14.4 µg/L and 19.2 µg/L. 

We further characterized the diagnostic performance of PIIINP with a prediction of the logistic model curve (Appendix A), a calibration curve (Appendix A) and a contingency table (Appendix A).

## 4. Discussion

To our knowledge, this is the first prospective study evaluating PINP and PIIINP as prognostic markers in severe ARDS patients treated with veno-venous ECMO. PIIINP was significantly higher in non-survivors than in survivors at all studied points in time during the course of treatment (Figure 2A). The concentration of serum PIIINP at the time of ECMO initiation was highly predictive of death during the course of treatment (Figure 2B and Table 2). Despite significant higher concentrations of serum PINP in non-survivors at the time of ECMO initiation and during the early course of treatment (Figure 1), PINP was not predictive for the clinical outcome.

Interestingly, the mortality prediction scores PRESERVE and RESP did not differ between survivors and non-survivors at the time of ECMO initiation and there was no difference in organ function and disease severity, measured by SAPS II score, SOFA score and APACHE II score (Table 1). These findings are in line with recent studies measuring the performance of mortality prediction scores for severe ARDS treated with ECMO where PRESERVE and RESP showed varying results compared to the initial publication [42,43].

### 4.1. Procollagen I and III in ARDS

Although non-survivors had significantly higher concentrations of serum PINP in the present study at the time of ECMO initiation it did not predict outcome. As the deposition of collagen I is part of the late profibrotic process in ARDS [16,17] the difference in serum concentration between resolving and unresolving ARDS at the time of ECMO initiation might be smaller compared to PIIINP.

Procollagen III-peptide has been studied in several studies in patients with ARDS and was associated with poor outcome [19,20,21,22,23,24,25,26]. The study by Meduri et al. detected higher concentrations of serum PIIINP in non-survivors compared to survivors on day 7 after ARDS onset [24]. Furthermore, serum PINP and PIIINP levels continued to increase exclusively in patients with unresolving ARDS. The predictive value of PIIINP in the present study is in line with observed difference between survivors and non-survivors on day 7 by Meduri et al. [24] considering the time of mechanical ventilation prior to ECMO treatment and study inclusion.

In our study, serum PIIINP was higher at all points in time in non-survivors compared to survivors. The baseline PIIINP at the time of ECMO initiation showed a high predictive value according to the AUC as well as a good calibration and goodness of fit according to the Brier score and LRT. Furthermore, the calculated DOR for PIIINP indicates a high diagnostic and discriminatory performance of the marker in our study cohort. 

Steinberg et al. showed lower procollagen III-peptide concentrations at the time of study inclusion in patients with ARDS who survived longer than 14 days [44]. The authors hypothesized that the lower baseline level of procollagen III-peptide might indicate less active fibroproliferation in these patients. 

Marshall et al. showed higher BAL concentrations of PIIINP in non-survivors compared to survivors and the same, non-significant trend for serum PIIINP [22]. A possible explanation for the observed significant differences in serum PIIINP at all points in time in the present study could be the severity of ARDS. All of the patients studied required veno-venous ECMO and the severity of illness, measured by SAPS II, SOFA and APACHE II score, was high compared to the previous studies investigating serial measurement of PINP and PIIINP [20,21,22,23,24] as well as other recent studies including ARDS patients treated with ECMO [42,43,45,46].

Forel et al. found significantly higher serum PIIINP concentrations in patients with unresolving ARDS who developed pulmonary fibroproliferation [18], which might progress to fibrosis and thus increase morbidity [47,48].

The results of the present study with significantly higher PIIINP serum concentrations in non-survivors are in accordance with the findings by Forel et al. given the fact that pulmonary fibroproliferation is a critical factor for determining the outcome of ARDS [22,49].

Forel et al. reported a serum PIIINP threshold of 16 µg/L to diagnose lung fibroproliferation with an AUC of 0.75 (95% CI 0.57–0.92) [18]. The AUC for serum PIIINP was not significantly smaller compared to the AUC reported for BAL PIIINP.

The threshold for serum PIIINP of 12.8 µg/L predicting the course of ARDS in the present study is comparable to the threshold calculated by Forel et al. and fibroproliferation might have been a major factor contributing to the mortality in our study. Furthermore, the median serum PIIINP concentration in survivors at all points in time during the course of treatment was lower than the reported threshold by Forel et al. to diagnose fibroproliferation [18].

Despite this correlation, it is important to differentiate between an adaptive and maladaptive fibroproliferation [50]. Fibroproliferation is an integral part of the tissue defense response and varies significantly between resolving and unresolving ARDS [27]. Unresolving ARDS is characterized by excessive, maladaptive fibroproliferation that is driven by persistent systemic inflammation [27]. The amount of systemic inflammation is crucial to determine if the fibroproliferative response leads to resolution of the lung injury or maladaptive lung repair with fibrosis occurs [27,50].

There is growing evidence that at least two subphenotypes of ARDS can be differentiated with distinct clinical features which might warrant a different treatment strategy [51,52,53,54,55]. The hyperinflammatory subphenotype is characterized by higher and persistent levels of inflammatory markers, fewer ventilator-free days and higher mortality. Furthermore, the hyperinflammatory subphenotype showed higher levels of markers for endothelial and epithelial lung injury [53]. The hypoinflammatory subphenotype can be characterized by less severe and decreasing inflammation, absence of shock, more ventilator-free days and overall lower mortality [51,52,53,54,55]. In the present study, inflammatory markers did not vary between survivors and non-survivors at the time of ECMO initiation. CRP as a marker of inflammation decreased exclusively in survivors between the time of ECMO initiation and day 5 as well as day 10, indicating a decreasing inflammatory stimulus (Appendix A). The decrease in PCT between the time of ECMO initiation and day 5 as well as day 10 in both groups suggests a sufficient antimicrobial therapy.

As a result of the persisting inflammatory stimulus and the unresolved hemodynamic impairment, the cumulative fluid balance in non-survivors was significantly higher during the course of treatment. This could either be caused by the underlying disease or might indicate an injurious ventilator strategy resulting in VILI and concomitant pulmonary fibroproliferation [56,57,58].

Therefore, the utilization of ECMO to reduce the proinflammatory stimulus of an injurious ventilator strategy with high driving pressures needs to be evaluated. PNIIIP might be an attractive parameter to guide the time point of ECMO initiation and the accompanying ventilator strategy.

### 4.2. ECMO Support and Respiratory Mechanics

At the time of ECMO initiation and during the early course of treatment, mechanical ventilation and ECMO support did not differ in both groups (Appendix A). In all patients, ventilator settings that are considered “ultra-protective” were used as described by recent studies [46,59].

Discontinuation of neuromuscular blockade and transition to an assisted-mode ventilation was only feasible in survivors, resulting in higher V_T_ and RR on day 10 compared to the time of ECMO initiation. Simultaneously, in survivors ECMO support could be reduced as indicated by the lower ECMO blood and gas flow on day 10 compared to the time of ECMO initiation. Another indicator of the clinical improvement in survivors might be the improved C_rs_ in survivors on day 3 during the course of treatment. This also results in a lower driving pressure in survivors on day 3 as the driving pressure derives from the ratio of tidal volume and C_rs_. An inverse relationship between C_rs_, which per se is strongly correlated with survival [60], and the concentration of PIIINP has been shown in patients with ARDS [61]. The higher positive fluid balance in non-survivors during the course of treatment might be discussed as another cause of the observed differences in C_rs_. A fluid management strategy targeting a negative fluid balance has been shown to shorten the duration of mechanical ventilation in ARDS patients [62].

Although we cannot exclude tidal hyperinflation [63] the pursued ventilator strategy for patients treated with ECMO in our study can be considered “ultra-protective” [64]. However, despite a substantial reduction of the driving pressure and the energy transmitted by the ventilator because of the utilization of ECMO, we were not able to reduce PIIINP in our observation period in non-survivor.

The interaction and time sequence between inflammation and pulmonary fibrosis is not clearly described. Experimental data by our group showed a reduced procollagen I and III RNA synthesis as well as less α-SMA expressing tissue in an animal model challenged with progredient higher LPS doses [65]. This is contrary to the findings of other groups, who reported an elevation in profibrotic parameters after extended periods of mechanical ventilation [66,67].

So as the incidence and evolution of pulmonary fibrosis in patients with ARDS is probably a complex interplay of inflammatory stimulus and ventilator-induced injury. It is unclear if profibrotic markers, such as PIIINP, are an independent predictor of mortality in our study but may integrate both influencing factors in a relevant way [68]. Therefore, PIIINP might be an attractive biological marker to individualize the ventilator strategy in ECMO patients.

Furthermore, the utilization of ECMO to reduce the proinflammatory stimulus of an injurious ventilator strategy with high driving pressures needs to be evaluated further, PNIIIP might be an attractive parameter to guide the time point of ECMO initiation and the accompanying ventilator strategy. Taken together the synthesis of profibrotic markers in patients with ARDS is influenced by the pulmonary inflammation of the underlying disease as well as the chosen ventilator strategy. As we describe a correlation between higher PIIINP and worse outcomes, and not a causality per se, we cannot differentiate whether elevated PIIINP is associated with injurious ventilation, inflammatory affection of the lungs or a deleterious combination of both factors. On the other hand, both of these injury mechanisms have been shown to influence mortality in patients with severe ARDS [2,60].

### 4.3. Limitations

The main limitation of the present study is the small sample size as only 23 patients could be included during the study period. This is due to the fact that only very severe cases of ARDS were studied and all patients that needed ECMO support other than veno-venous were excluded. In the present study, severity of illness at the time of ECMO initiation, indicated by prediction scores and organ failure scores, and duration of mechanical ventilation prior to inclusion did not differ between both groups hence making a selection bias unlikely. Furthermore, the studies by Marshall et al. [22], Meduri et al. [24] and Forel et al. [18] had comparable sample sizes and yielded significant results as well.

We acknowledge that the reported diagnostic values for PIIINP may not be precise and presumably overstated due to the small sample size and might vary in future studies. In addition, the assessment of model calibration may be considerably influenced by the small sample size and the inability to adjust for baseline characteristics. Therefore, our data interpretation is solely hypothesis generating and needs confirmation in a prospective clinical trial.

We found a non-significant difference of approximately two days in mechanical ventilation prior to transfer to ECMO therapy that might have an influence on PIIINP levels at the time of ECMO initiation, especially as the timing of pulmonary fibroproliferation is characterized [69,70,71] and might be modulated by inflammation and the ventilator strategy. On the other hand, we found no significant differences of PIIINP in survivors when comparing day 0 and day 3. Furthermore, there was a significant difference in PIIINP between survivors and non-survivors even at day 10. So, we hypothesize that there has to be a constellation of findings in non-survivors initiating pulmonary fibroproliferation and ECMO per se is not able to reverse that although driving pressure and the mechanical power transferred by the ventilator are markedly reduced compared to a conservative treatment strategy.

Although we a priori excluded patients with end-stage chronic organ failure and non-pulmonary fibrotic diseases, there is a possibility that non-recognized tissue fibrosis might interfere with the pulmonary procollagen accumulation. However, because of the high percentage (87%) of pulmonary ARDS in the study population, the measured differences in procollagen serum concentration are most likely the result of pulmonary procollagen synthesis.

We measured PINP and PIIINP concentrations in serum instead of in a BAL because of the simple and non-invasive availability, the high reliability and the validation of the used assays for serum measurement. Of course, this might confound our results because of the typical alteration of the alveolar capillary membrane in ARDS patients. We cannot rule out a compartmentalization of fibroproliferation in the alveolar space without any spillover of PINP or PIIINP in the pulmonary circulation. On the other hand, there are comparable data from serum samples in previous studies [18,24], which are in line with our results.

We only studied one protein from a complex biological system in a disease with significant heterogeneity hence procollagen liberation may stem from physiological tissue repair or excessive, maladaptive fibroproliferation. To fully understand the kinetics of procollagen deposition and validate PINP or PIIINP as a marker to guide treatment, more prospective studies are needed.

## 5. Conclusions

In ARDS patients treated with ECMO, serum PINP and PIIINP concentrations differ between survivors and non-survivors. The concentration of PIIINP at the time of ECMO initiation might be associated with a worse outcome.

Furthermore, PIIINP might be a marker to detect patients with high risk for unresolving ARDS and poor outcomes. The use of PIIINP to initiate and guide an individualized treatment should be evaluated in future trials.

## Figures and Tables

**Figure 1 jcm-10-03686-f001:**
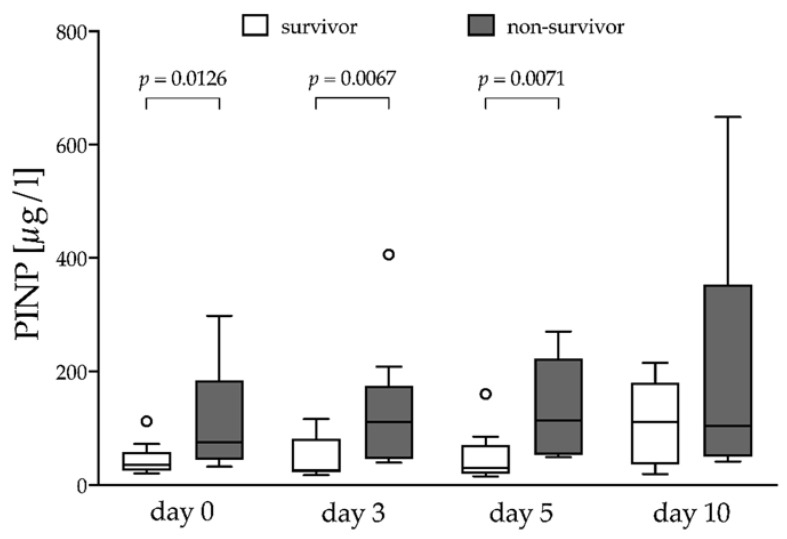
N-terminal procollagen I-peptide. Survivor vs. non-survivor, box plots using the Tukey method; PINP, N-terminal procollagen I-peptide. Outliers are marked as circles.

**Figure 2 jcm-10-03686-f002:**
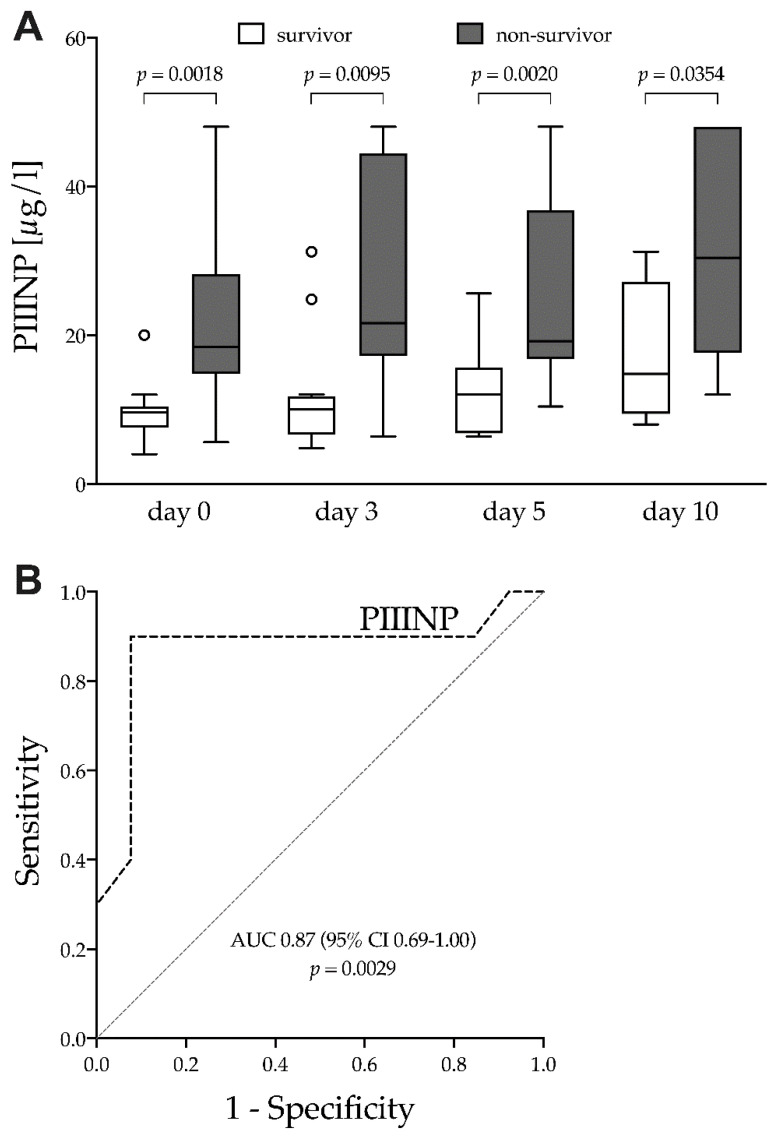
N-terminal procollagen III-peptide. (**A**) survivor vs. non-survivor, box plots using the Tukey method; (**B**) ROC curve for PIIINP at the time of ECMO initiation, PIIINP, N-terminal procollagen III-peptide; AUC, area under the ROC curve; ROC, Receiver operating characteristic; ECMO, extracorporeal membrane oxygenation. Outliers are marked as circles.

**Table 1 jcm-10-03686-t001:** Characteristics of the patients.

	ICU Survivor (n = 13)	ICU Non-Survivor (n = 10)	*p*-Value
Sex (male)	11 (85%)	4 (40%)	0.0393
Age (years)	54.2 ± 9.2	59.7 ± 9.4	0.1695
Height (cm)	177.7 ± 9.6	171.8 ± 8.9	0.1477
Body weight (kg)	104.5 ± 17.6	95.4 ± 19.6	0.2569
Body mass index (kg/m^2^)	33.5 ± 7.6	32.7 ± 8.5	0.8262
Main cause of ARDS (n (%))			
Pneumonia	12 (92)	7 (70)	0.5596
Aspiration	0 (0)	1 (10)	0.4348
Extrapulmonary-sepsis	1 (8)	2 (20)	0.5596
SAPS II	67.4 ± 14.0	72.4 ± 6.6	0.3096
SOFA	13.3 ± 3.6	15.5 ± 2.7	0.1265
APACHE II	28.7 ± 8.5	31.9 ± 4.1	0.2852
RESP	−4.0 ± 4.3	−5.9 ± 4.7	0.3256
PRESERVE	5.1 ± 2.2	5.6 ± 2.2	0.5773
MV prior to inclusion (days)	4.5 ± 2.7	6.9 ± 4.1	0.1415
Transferred from external hospital	5/13	6/10	0.8790
Salvage ECMO	2/13	3/10	0.7443
Driving pressure before ECMO	18.4 ± 5.3	17.7 ± 4.2	0.2891
P/F-ratio prior to ECMO	87.1 ± 23.6	79.1 ± 27.8	0.3564
Prone positioning before ECMO	6/13	6/10	0.1459
Neuromuscular blocking agents before ECMO	11/13	10/10	0.8995
Dialysis/CRRT before ECMO	3/13	2/10	0.4378
Duration of ECMO support (days)	11.2 ± 2.9	16.4 ± 8.1	0.0437
ICU length of stay (days)	37.0 (32–73)	24.0 (15.8–33.5)	0.0039

Data are *n* (%), mean ± standard deviation or median (interquartile range). ARDS, acute respiratory distress syndrome; SAPS II, Simplified Acute Physiology Score II; SOFA, Sequential Organ Failure Assessment; APACHE II, Acute Physiology And Chronic Health Evaluation II; RESP, Respiratory ECMO Survival Prediction; PRESERVE, Predicting Death for Severe ARDS on vv-ECMO; MV, mechanical ventilation; ECMO, extracorporeal membrane oxygenation; P/F ratio, ratio between the arterial oxygen partial pressure and the fraction of inspired oxygen; CRRT, continuous renal replacement therapy; ICU, intensive care unit.

**Table 2 jcm-10-03686-t002:** Diagnostic performance of serum N-terminal procollagen I-peptide and N-terminal procollagen III-peptide.

	Sensitivity (%)	Specificity (%)	Positive Predictive Value (%)	Negative Predictive Value (%)	Likelihood Ratio Positive	Likelihood Ratio Negative	Diagnostic Odds Ratio
PIIINP >12.8 µg/L	90.0 (59.6–99.5)	92.3 (66.7–99.6)	90.0 (59.6–99.5)	92.3 (66.7–99.6)	11.7 (1.7–77.8)	0.1 (0.02–0.7)	108 (5.9–1969.5)

Data are % (95% CI) or absolute values (95% CI); PIIINP, N-terminal procollagen III-peptide.

**Table 3 jcm-10-03686-t003:** Diagnostic performance of serum N-terminal procollagen III-peptide at different cut-offs.

	Number of Patients	TP	TN	FP	FN	Sensitivity (%)	Specificity (%)	PPV (%)	NPV (%)	Diagnostic Odds Ratio
PIIINP >8.4 µg/L	18	9	4	9	1	90.0 (59.6–99.5)	30.7 (12.7–57.6)	50.0 (39.7–60.3)	34.4 (34.5–76.8)	4.0 (0.4–43.1)
PIIINP >12.8 µg/L	10	9	12	1	1	90.0 (59.6–99.5)	92.3 (66.7–99.6)	90.0 (59.6–99.5)	92.3 (66.7–99.6)	108 (5.9–1969.5)
PIIINP >14.4 µg/L	9	8	12	1	2	80.0 (49.0–96.5)	92.3 (66.7–99.6)	88.9 (54.3–98.2)	85.7 (63.2–95.4)	48.0 (3.7–622.0)
PIIINP >19.2 µg/L	5	4	12	1	6	40.0 (16.8–68.7)	92.3 (66.7–99.6)	80.0 (34.4–96.8)	66.7 (54.1–77.3)	8.0 (0.7–88.2)

TP, true positive; TN, true negative; FP, false positive; FN, false negative; PPV, positive predictive value; NPV, negative predictive value.

## Data Availability

The datasets analyzed during the current study are available from the corresponding author on reasonable request.

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
