# Peer review of "Procollagen I and III as Prognostic Markers in Patients Treated with Extracorporeal Membrane Oxygenation: A Prospective Observational Study"

_jcm, 2021, doi:10.3390/jcm10163686_

Round 1

Reviewer 1 Report

Dear Authors

I appreciated your research, because I agree that lung fibroproliferation in ARDS could be very detrimental for patients and their outcome. Your effort to identify an early biological marker of this process as PIIINP could be very useful in clinical practice.

Before editing your draft, I've only some suggestions and queries for you in attached file

Best regards

Author Response

Response to Reviewers’ Comments 

Reviewer 1

Concerns:

1.: Pag. 2 line 47: better explain if you follow a protocol of mechanical respiratory assesment before starting assisted ventilation in ECMO. In my opinion, if driving pressure is still high and Crs low, as in “non survivors” (tab 1), an early assisted ventilation could be unsafe even in ECMO.

Re.: We thank the reviewer for this valuable comment and fully agree that reviewer that a high respiratory drive and transpulmonary pressures are a common clinical problem in patients treated with ECMO during assisted spontaneous breathing [1]. We further elaborated on the standard operating procedure in our unit for this important topic in the manuscript. Generally, we try to avoid high driving pressures because of vigorous spontaneous breathing and titrate ECMO support, sedatives and neuromuscular blocking agents accordingly (page 3, line 13-24).

2.: Tab. 2 pag 5: it’s not clear for me to understand driving pressure’s trend between “survivors” and “non-survivors”. In effect is predictable to find higher value in “non-suvivors2 at day 3, but unexpected a lower one at day 10. The same claim for compliance’s trend: lower in “non survivors” at day 3, equal at day 10.

Re.: As per our standard operation procedure of our unit patients on vvECMO support initially receive neuromuscular blockade and are ventilated with reduced respiratory rates and tidal volumes resulting in a marked reduced driving pressure. We further described the protocol in the material and methods section (please see page 3, line 13-24).

In the early phase of ECMO support we aim for a negative fluid balance and to wean the noradrenaline support if the patient is hemodynamically stable. If these prerequisites are met, we discontinue neuromuscular blockade, reduce sedative agents and establish an assisted ventilator mode with lung protective settings. As shown in Supplemental table 1 typically we did not reach a negative fluid balance in the non-survivor group possibly resulting in an elevated driving pressure at day 3 and a reduced compliance of the respiratory system at day 3. On the other hand, at day 10 survivors typically were ventilated in an assisted ventilator mode contrary to survivors who were typically still ventilated in the protective controlled mode. The assisted spontaneous breathing of the survivors therefore results in higher (but still considered “safe”) driving pressures at day 10. We elaborated on that in the Discussion (please see page 11, line 1-8). Please not that we shifted table 2 in the supplement as recommended by reviewer 2.

3.: Pag 10 line 40: You report at day 3 similar values of ECMO sweep gas flow, TV and PaCO2 in two patients’ groups. That could mean an increased risk of VILI even in ECMO for “non survivors”. In other terms an increased lung’s fibroproliferative risk, due to their high driving pressure. Elevated serum PIIINP could be one of the most important biological marker of this process.

Re.: We fully agree with the reviewer that PIIINP might be an important biological marker for the interpretation of driving pressure for the risk of excessive fibroproliferation in the individual patient, even if they are in a range generally considered “safe” [2]. This correlation has been shown experimentally already [3, 4]. On the other hand, the effects of proinflammatory stimuli regarding the effects on pulmonary fibrosis are not well defined.  So in our study, we cannot differentiate whether elevated PIIINP is associated with injurious ventilation, inflammatory affection of the lungs or a deleterious combination of both factors.

We elaborated on this concept in the manuscript (page 11, line 15-41).

4.: Pag. 11 line 5: I think that the relevant fluid balance difference could related to a different septic condictions in patients. My assumption is based on PCT value at day 0 (20 in “non survivors”, vs 4 in “survivors”). The PCT decrease reveals your good antimicrobial strategy, but this difference could also explain the high mortality and need for fluids in “non-survivors” group, even if SAPS II and SOFA scores don’t differ.

Re.: As indicated by reviewer, the higher procalcitonin at day 0 and the cumulative positive fluid balance in non-survivors might indicate an ongoing proinflammatory stimulus that promotes fibroproliferation in the injured lungs. This could either be caused by the underlying disease or might indicate an injurious ventilator strategy resulting in VILI and concomitant pulmonary fibroproliferation. We incorporated this aspect in the manuscript (page 10, line 39-41)

5.: Pag. 11 line 29: PIIINP could also suggest and drive a new ventilatory-strategy during ECMO for reducing VILI, in particular in patients with elevated infection’s risk.

Re.: We fully agree with the reviewer that PIIINP might be useful to evaluate ventilator strategies regarding there biological impact on fibroproliferation and concomitant ventilator induced lung injury due to inhomogenous stress distribution in the lung parenchyma.  On the other hand, we think that the characteristics of PIIINP need to be better described in relation to the amount of inflammation caused either by the underlying disease or the ventilator. Our group found a negative correlation between Interleukin 6 and PIIINP during the early phases of injurious ventilation [5-7] associated with higher driving pressures. So, we suspect that the duration of the ventilator strategy, the specific condition of the patient and the underlying disease are important influencing factors for the generation of PIIINP and its effects on the lung parenchyma. 

The reduction of the inflammatory and the corresponding fibroproliferative stimulus from the ventilator due to the utilization of ECMO needs to be evaluated in further studies. We briefly elaborated on the concept in the manuscript (page 10, line 42-45 and page 11, line 19-44).

Reviewer 2 Report

            Boesing and colleagues have reported the findings of a prospective cohort study examining the predictive ability of N-terminal procollagen I- and III-peptide (PINP and PIIINP) in patients with severe ARDS who received treated with extracorporeal membrane oxygenation (ECMO). Through prospective data collection the authors were able to obtain detailed granular data on all patients in the study. The strengths of this study are the granular data and the potential of a novel prognostic markers in ARDS. Using this data, the authors demonstrated that elevated levels of PIIINP are associated with an increased odds of death while receiving ECMO.

            I would like to thank the authors for the opportunity to read their paper and hope that my comments will be of help.

Main concerns:

  1. I am unclear of the significance of the outcome of interest, survival to decannulation. I would suggest the author’s use a more patient-centered outcome such as survival to hospital discharge. Acknowledging the nature of this study and the focus on respiratory mechanics could consider survival to ICU discharge, balancing the prediction of recovery of pulmonary function and the import of the outcome to the patient. 

  1. I find the results section hard to follow and would recommend the authors include additional statistical analyses related to predictive studies, such as a calibration curve.

  1. The initial sentence in the conclusion “the concentration of PIIINP at the time of ECMO initiation is predictive of death during the course of treatment” overstates the significance of the findings. As the authors state in their discussion this research is hypothesis generating. At most this study suggests the association between elevated PIIINP concentrations and worse outcomes.

Specific suggestions/comments:

Abstract:

  1. Recommend including a description of the study design in the methods section of the abstract (prospective cohort study of patients with severe ARDS who received ECMO).
  2. Also consider highlighting that a univariable logistic regression model was utilized.
  3. In results section consider adding the number of survivors/deaths.
  4. As stated above the conclusion appears to overstate the findings. Would recommend the authors reword to better highlight the exploratory nature of this work.

Methods:

  1. Consider further subdividing the methods section for reader ease, i.e. study design, participants, standard therapy, data collection, statistical analysis
  2. Recommend adding further details to patient inclusion: was severe ARDS identified using the Berlin criteria, was there a minimum or maximum number of days of ventilation prior to ECMO
  3. Recommend additional statistical analysis related to prognostic modelling including a calibration curve and Brier-Score. I find this article a clear example of modelling used for predictive tools (doi: https://doi.org/10.1136/bmj.m3339)

Results:

  1. If possible, I would recommend the authors add additional data to table 1 including: tidal volume prior to ECMO, PF ratio prior to ECMO, receipt of prone ventilation, receipt of NMB prior to ECMO, receipt of dialysis (if this occurred). Ideally, would like to make it clear to the readers that these patients were equally ill and also received the same treatment.
  2. Although not statistically significant, the difference of 2 days in the duration of ventilation prior to ECMO does standout, especially if mechanical ventilation is injurious and causing increased PIIINP levels. Although this relationship would not change the findings of the study (increased PIIINP is associated with worse outcomes) would recommend the authors address the clinical difference in duration of ventilation in the discussion.
  3. I am unsure of the significance of the data related to the treatment while on ECMO. Unless the authors detected major treatment differences between groups it does not significantly add to an analysis of a predictive model. I would recommend the authors shorten this section to a sentence or two simply stating that patients received the same treatment and then moving tables 2 and 3 to a supplement.
    1. If the authors retain table 2 in the primary manuscript recommend simplifying the table, it is very overwhelming.
  4. The authors have identified that PINP is not associated with death but there is an AUC of 0.80. Many readers unfamiliar with predictive modelling may find this discrepancy confusing. If the authors wish to present the AUC of a non-significant model, which I am unsure of the value, it is likely important to highlight what these discrepant results indicate, i.e. in randomly chosen patients most of the time the one who dies will have a higher PINP but an higher PINP is not associated with death.
  5. Recommend including some assessment of calibration in the results, if the authors remove tables 2 and 3 (and the associated text) this will provide additional space for an evaluation of calibration.

Discussion:

  1. As mentioned above although the difference in duration of ventilation is not statistically significant, this finding is likely related to the sample size as a difference of 2.5 days appears clinically significant. Would recommend the authors further explore this possibility.
  2. As the goal of this paper is a discussion of the predictive value of PINP and PIIINP levels recommend the authors focus their discussion on these areas as opposed to discussing support during ECMO across groups.
  3. Given the lack of ability to adjust for other factors I think many readers may be unsure if PIIINP is a true predictor or associated with other already measured factors (duration of ventilation, inflammation, etc). Are the authors able to further elaborate on why these biomarkers may be of import.
  4. The authors have acknowledged the sample size but would recommend further exploration of the limitation, including that the predictive ability of these markers may be overstated is this study, and an inability to adjust for baseline characteristics.

Conclusion:

  1. The initial sentence in the conclusion “the concentration of PIIINP at the time of ECMO initiation is predictive of death during the course of treatment” overstates the significance of the findings. As the authors state in their discussion this research is hypothesis generating. At most this study suggests the association between elevated PIIINP concentrations and worse outcomes.

Minor comments:

Introduction:

  1. Could benefit from review of sentence structure, for example the sentence at line 34 “Up to date, no pharmacologic treatment has been shown to reduce mortality.” is awkwardly worded.

Methods:

  1. Consider specifying why certain patient groups were excluded, particularly pregnant patients.

Results:

  1. I am uncertain of the value of including diagnostic accuracy in table 4. Given how susceptible accuracy is to prevalence it is not a great measure of predictive value. If the authors are interested in a single summary statistic consider using the diagnostic odds ratio (DOI: 1016/s0895-4356(03)00177-x)
  2. Consider displaying the diagnostic performance of PINP and PIIINP at a variety of cut-off ratios. Doing so can help the reader identify a cut-off that may be better for their particular application.

Again, I would like to thank the authors for the opportunity to read their manuscript.

Author Response

Reviewer 2

Boesing and colleagues have reported the findings of a prospective cohort study examining the predictive ability of N-terminal procollagen I- and III-peptide (PINP and PIIINP) in patients with severe ARDS who received treated with extracorporeal membrane oxygenation (ECMO). Through prospective data collection the authors were able to obtain detailed granular data on all patients in the study. The strengths of this study are the granular data and the potential of a novel prognostic markers in ARDS. Using this data, the authors demonstrated that elevated levels of PIIINP are associated with an increased odds of death while receiving ECMO.

I would like to thank the authors for the opportunity to read their paper and hope that my comments will be of help.

Re.: We thank the reviewer for his positive comment and want to emphasise that in our opinion his comments were very helpful to improve the manuscript.

Concerns:

Main concerns:

1.: I am unclear of the significance of the outcome of interest, survival to decannulation. I would suggest the author’s use a more patient-centered outcome such as survival to hospital discharge. Acknowledging the nature of this study and the focus on respiratory mechanics could consider survival to ICU discharge, balancing the prediction of recovery of pulmonary function and the import of the outcome to the patient.

Re.: We thank the reviewer and fully acknowledge that the metric survival to hospital discharge is a valuable information and might help to interpret the amount of fibroproliferation due to pulmonary inflammation and the functional outcome of the patient. We therefore included this in the results (page 4, line 33-34). On the other hand, we want to emphasize that in this study we aimed to explore the association of profibrotic markers and ICU mortality analogous to Forel et al. in a circumscribed population with most severe ARDS and based our analyses accordingly on the differentiation of ICU survivors and non-survivors [8]. We precised this fact in the results section and in table 1.

2.: I find the results section hard to follow and would recommend the authors include additional statistical analyses related to predictive studies, such as a calibration curve.

Re.: We restructured the statistical analysis section according to the recommendations of the reviewer (page 4, line 6-27). We now report the Brier-Score for PIIINP as well as the diagnostic odds ratio in table 2. We further present a table containing the diagnostic performance of serum N-terminal procollagen III-peptide at a cut-off ratio of > 8.4 µg/l, > 12.8 µg/l, > 14.4 µg/l and 19.2 µg/l (table 3) and a calibration curve of the model (Supplemental Figure 2) in the manuscript. Furthermore, we added a curve showing the prediction of the logistic model (Supplemental figure 1) and a contingency table of serum PIIINP at a cut-off of > 12.8 µg/l in the supplement (Supplemental table 3).

3.: The initial sentence in the conclusion “the concentration of PIIINP at the time of ECMO initiation is predictive of death during the course of treatment” overstates the significance of the findings. As the authors state in their discussion this research is hypothesis generating. At most this study suggests the association between elevated PIIINP concentrations and worse outcomes.

Re.: We reworded the sentence according to the recommendations of the reviewer (page 12, line 37-38).

Specific suggestions/comments:

Abstract:

4.: Recommend including a description of the study design in the methods section of the abstract (prospective cohort study of patients with severe ARDS who received ECMO).

Re.: We reworded the sentence according to the recommendations of the reviewer (page 1, line 16-19).

5.: Also consider highlighting that a univariable logistic regression model was utilized.

Re.: According to the recommendations of the reviewer, we stated the statistical model more precisely (page 1, line 19-21).

6.: In results section consider adding the number of survivors/deaths.

Re.: We added the number of survivors to ICU discharge in the results section (page 1, line 21)

7.: As stated above the conclusion appears to overstate the findings. Would recommend the authors reword to better highlight the exploratory nature of this work.

Re.: We stated the exploratory hypothesis generating nature of the study more precisely in the conclusion section of the abstract (page 1, line 26-28).

Methods:

8.: Consider further subdividing the methods section for reader ease, i.e. study design, participants, standard therapy, data collection, statistical analysis

Re.: We restructured the methods section according to the recommendations of the reviewer.

9.: Recommend adding further details to patient inclusion: was severe ARDS identified using the Berlin criteria, was there a minimum or maximum number of days of ventilation prior to ECMO

Re.: We thank the reviewer for this valuable comment and precised our standard operation procedure for the conservative management of severe ARDS and prior to ECMO cannulation (page 2, line 28-35 and line 40-48).

10.: Recommend additional statistical analysis related to prognostic modelling including a calibration curve and Brier-Score. I find this article a clear example of modelling used for predictive tools (doi: https://doi.org/10.1136/bmj.m3339)

Re.: Following the recommendations of the reviewer, restructured the description of the statistical analysis (page 4, line 6-27) and included the calibration curve (Supplemental Figure 2) and the Brier-score (page 6, line 11) in the results.

Results:

11: If possible, I would recommend the authors add additional data to table 1 including: tidal volume prior to ECMO, PF ratio prior to ECMO, receipt of prone ventilation, receipt of NMB prior to ECMO, receipt of dialysis (if this occurred). Ideally, would like to make it clear to the readers that these patients were equally ill and also received the same treatment.

Re.: We thank the reviewer for this valuable comment and integrated the proposed information in table 1.

12.: Although not statistically significant, the difference of 2 days in the duration of ventilation prior to ECMO does standout, especially if mechanical ventilation is injurious and causing increased PIIINP levels. Although this relationship would not change the findings of the study (increased PIIINP is associated with worse outcomes) would recommend the authors address the clinical difference in duration of ventilation in the discussion.

Re.: We fully agree with the reviewer, that the difference of approximately two days in mechanical ventilation prior to transfer to ECMO therapy might have an influence on PIIINP levels at the time of ECMO initiation, especially as the timing of pulmonary fibroproliferation is not characterized yet [5, 7, 9, 10] and might be modulated by inflammation and the ventilator strategy. On the other hand, we found no significant differences of PIIINP (1.5 ± 0.5 µg/l vs. 1.5 ± 1.0 µg/l, p = 0.232) in survivors comparing day 0 and day 3 suggesting another causative element besides the applied ventilator/ECMO strategy and the inflammatory stimulus in non-survivors. Furthermore, there was a significant difference in PIIINP between survivors and non-survivors even at day 10. So, we hypothesize that there has to be a constellation of findings in non-survivors initiating pulmonary fibroproliferation and our treatment strategy with ECMO, resulting in reduced driving pressure and mechanical power transferred by the ventilator, is not able to influence the increased production of PIIINP in non-survivors. We briefly discussed this in the manuscript (page 12, line 7-16).

13.: I am unsure of the significance of the data related to the treatment while on ECMO. Unless the authors detected major treatment differences between groups it does not significantly add to an analysis of a predictive model. I would recommend the authors shorten this section to a sentence or two simply stating that patients received the same treatment and then moving tables 2 and 3 to a supplement.

Re.: Following the recommendations of the reviewer, we integrated table 2 and 3 and the accompanying results in the supplement.

14.: If the authors retain table 2 in the primary manuscript recommend simplifying the table, it is very overwhelming.

Re.: see point 13.

15.: The authors have identified that PINP is not associated with death but there is an AUC of 0.80. Many readers unfamiliar with predictive modelling may find this discrepancy confusing. If the authors wish to present the AUC of a non-significant model, which I am unsure of the value, it is likely important to highlight what these discrepant results indicate, i.e. in randomly chosen patients most of the time the one who dies will have a higher PINP but an higher PINP is not associated with death.

Re.: We thank the reviewer for this valuable comment. We used AUC to estimate the discriminative value of our logistic model because of its widespread use in the recent literature and to allow comparisons between predictive studies. We are aware of the drawbacks of the AUC alone, we therefore removed the AUC of the non-significant model in Figure 1 and Table 2. Furthermore, we modified Table 2 and 3, Figure 3 and Supplemental figure 1 and Table 3.

16.: Recommend including some assessment of calibration in the results, if the authors remove tables 2 and 3 (and the associated text) this will provide additional space for an evaluation of calibration.

Re.: Following the recommendations of the reviewer, we included the Brier-score to allow for an evaluation of calibration and overall goodness of fit (page 6, line 11). Furthermore, we used the likelihood ratio test to determine the fit of the logistic model (page 6, line 11) and added a calibration curve to the supplement (Supplemental Figure 2).

Discussion:

17.: As mentioned above although the difference in duration of ventilation is not statistically significant, this finding is likely related to the sample size as a difference of 2.5 days appears clinically significant. Would recommend the authors further explore this possibility.

Re.: We included a brief discussion of the relevance of the different days of MV prior to ECMO in the limitations (please also see point 12; page 12, line 7-16).

18.: As the goal of this paper is a discussion of the predictive value of PINP and PIIINP levels recommend the authors focus their discussion on these areas as opposed to discussing support during ECMO across groups.

Re.: According to the recommendations of the reviewer, we shortened and rearranged the discussion and integrated the relevant parts of 4.3 in 4.1 and briefly commented on the Brier score and the diagnostic odds ratio in the discussion (page 9, line 33-36)

19.: Given the lack of ability to adjust for other factors I think many readers may be unsure if PIIINP is a true predictor or associated with other already measured factors (duration of ventilation, inflammation, etc). Are the authors able to further elaborate on why these biomarkers may be of import.

Re.: As the synthesis of profibrotic markers in patients with ARDS is influenced by the pulmonary inflammation of the underlying disease as well as the chosen ventilator strategy, we fully agree with the reviewer, that we describe a correlation between higher PCIIINP and worse outcome and not a causality per se. On the other hand, both of these factors have been shown to influence mortality in these patients [2, 11]. Therefore, we would like to emphasize two points.

  • Although we cannot exclude tidal hyperinflation [12] the pursued ventilator strategy for patients treated with ECMO in our study can be considered “ultra-protective” [13]. So, despite a substantial reduction of the energy transmitted by the ventilator, we were not able to reduce PIIINP in our observation period in non-survivor.  
  • The interaction and time sequence between inflammation and pulmonary fibrosis is not clearly described. Experimental data by our group showed a reduced procollagen I and III RNA synthesis as well as less α-SMA expressing tissue in an animal model challenged with progredient higher LPS doses [7]. This is contrary to the findings of other groups [14, 15].

So as the incidence and evolution of pulmonary fibrosis in patients with ARDS is probably a complex interplay of inflammatory stimulus and ventilator-induced injury. It is unclear if profibrotic markers like PIIINP are an independent predictor of mortality in our study. On the other hand, we hypothesize, that PIINP integrates both influencing factors (inflammation, ventilator-induced lung injury) and thus may have interesting predictive properties.  We briefly integrated this line of thought in the discussion (page 11, line 15-41).

20.: The authors have acknowledged the sample size but would recommend further exploration of the limitation, including that the predictive ability of these markers may be overstated is this study, and an inability to adjust for baseline characteristics.

Re.: We thank the reviewer for this valuable comment. We emphasized the effect of the small sample size on the diagnostic performance measures and that the predictive ability of the marker may vary substantially in a larger patient cohort. (page 12, line 1-6)

Conclusion:

21.: The initial sentence in the conclusion “the concentration of PIIINP at the time of ECMO initiation is predictive of death during the course of treatment” overstates the significance of the findings. As the authors state in their discussion this research is hypothesis generating. At most this study suggests the association between elevated PIIINP concentrations and worse outcomes.

Re.: We reworded the sentence according to the recommendations of the reviewer (page 12, line 36-41)

Minor comments:

Introduction:

22.: Could benefit from review of sentence structure, for example the sentence at line 34 “Up to date, no pharmacologic treatment has been shown to reduce mortality.” is awkwardly worded.

Re.: We revised the manuscript according to the recommendations of the reviewer (page 1, line 34-38, page 2, line 8-10)

Methods:

23.: Consider specifying why certain patient groups were excluded, particularly pregnant patients.

Re.: We specified the exclusion criteria as recommended by the reviewer. Briefly, the exclusion of patients younger than 18 years and pregnant patients was required by the ethics committee. We elaborated on that in materials and methods (page 2, line 28-35).

Results:

24.: I am uncertain of the value of including diagnostic accuracy in table 4. Given how susceptible accuracy is to prevalence it is not a great measure of predictive value. If the authors are interested in a single summary statistic consider using the diagnostic odds ratio (DOI: 1016/s0895-4356(03)00177-x)

Re.: We now provide the diagnostic odds ratio in Table 2 and Table 3 according to the recommendations of the reviewer in table 2.

25.: Consider displaying the diagnostic performance of PINP and PIIINP at a variety of cut-off ratios. Doing so can help the reader identify a cut-off that may be better for their particular application.

Re.: We included a table displaying the diagnostic performance of PIIINP at a variety of cut-off ratios according to the recommendations of the reviewer in Table 3.

Again, I would like to thank the authors for the opportunity to read their manuscript.

Re.: We would like the reviewer for his thorough work. We feel it helps to improve the manuscript substantially.

References:

  1. Mauri, T., et al., Control of Respiratory Drive and Effort in Extracorporeal Membrane Oxygenation Patients Recovering from Severe Acute Respiratory Distress Syndrome. Anesthesiology, 2016. 125(1): p. 159-67.
  2. Amato, M.B., et al., Driving pressure and survival in the acute respiratory distress syndrome. N Engl J Med, 2015. 372(8): p. 747-55.
  3. Santos, C.L., et al., Effects of pressure support and pressure-controlled ventilation on lung damage in a model of mild extrapulmonary acute lung injury with intra-abdominal hypertension. PLoS One, 2017. 12(5): p. e0178207.
  4. Pinto, E.F., et al., Static and Dynamic Transpulmonary Driving Pressures Affect Lung and Diaphragm Injury during Pressure-controlled versus Pressure-support Ventilation in Experimental Mild Lung Injury in Rats. Anesthesiology, 2020. 132(2): p. 307-320.
  5. Krebs, J., et al., Time course of lung inflammatory and fibrogenic responses during protective mechanical ventilation in healthy rats. Respir Physiol Neurobiol, 2011. 178(2): p. 323-8.
  6. Krebs, J., et al., Open lung approach associated with high-frequency oscillatory or low tidal volume mechanical ventilation improves respiratory function and minimizes lung injury in healthy and injured rats. Crit Care, 2010. 14(5): p. R183.
  7. Krebs, J., et al., Effects of lipopolysaccharide-induced inflammation on initial lung fibrosis during open-lung mechanical ventilation in rats. Respir Physiol Neurobiol, 2015. 212-214: p. 25-32.
  8. Forel, J.M., et al., Type III procollagen is a reliable marker of ARDS-associated lung fibroproliferation. Intensive Care Med, 2015. 41(1): p. 1-11.
  9. Coalson, J.J., et al., Neonatal chronic lung disease in extremely immature baboons. Am J Respir Crit Care Med, 1999. 160(4): p. 1333-46.
  10. Yoder, B.A., et al., High-frequency oscillatory ventilation: effects on lung function, mechanics, and airway cytokines in the immature baboon model for neonatal chronic lung disease. Am J Respir Crit Care Med, 2000. 162(5): p. 1867-76.
  11. Bellani, G., et al., Epidemiology, Patterns of Care, and Mortality for Patients With Acute Respiratory Distress Syndrome in Intensive Care Units in 50 Countries. JAMA, 2016. 315(8): p. 788-800.
  12. Terragni, P.P., et al., Tidal hyperinflation during low tidal volume ventilation in acute respiratory distress syndrome. Am J Respir Crit Care Med, 2007. 175(2): p. 160-6.
  13. Rozencwajg, S., et al., Ultra-Protective Ventilation Reduces Biotrauma in Patients on Venovenous Extracorporeal Membrane Oxygenation for Severe Acute Respiratory Distress Syndrome. Crit Care Med, 2019. 47(11): p. 1505-1512.
  14. Domenici, L., et al., Evolution of endotoxin-induced lung injury in the rat beyond the acute phase. Pathobiology, 2004. 71(2): p. 59-69.
  15. Keshari, R.S., et al., Acute lung injury and fibrosis in a baboon model of Escherichia coli sepsis. Am J Respir Cell Mol Biol, 2014. 50(2): p. 439-50.

Round 2

Reviewer 2 Report

I would like to thank the authors for their detailed response to the review and the thorough changes they had made to their manuscript. I look forward to where this research leads in the future.